

# Cropping with *Vicia villosa* and native grass improves soil's bacterial structure and ecological network in a jujube orchard

Shoule Wang[1,2], Zhongtang Wang[1], Qing Hao[2], Bin Peng[3], Pan Li[1], Xuelong Qi[1] and Qiong Zhang[1]

[1] Shandong Institute of Pomology, Shandong Academy of Agricultural Sciences, Tai'a, China
[2] Research Institute of Horticultural Crops, Xinjiang Academy of Agricultural Sciences, Urumgi, China
[3] Xinjiang Institute of Ecology and Geography, Chinese Academy of Sciences, Urumgi, China

## ABSTRACT

In a jujube orchard, cropping withgrass may influence bacterial diversity and ecological networks due to changes of physicochemical properties in soil, which has a serious effect on the stability of soil ecosystems. The aim of this study was to analyze the effects of different cultivation methods (CK: cleaning tillage; NG: cropping with native grass; VV: cropping with *Vicia villosa*) on the soil's bacterial structure and its co-occurrence network in a jujube orchard. The results showed that the highest moisture content, total nitrogen, and organic matter in the rhizosphere soil of a jujube orchard was found in the VV group. The soil's moisture content, total nitrogen, and organic matter in the VV group were 2.66%, 0.87 g kg$^{-1}$, and 5.55 mg kg$^{-1}$ higher than that found in the CK group. Compared to the CK group, the number of unique species in the rhizosphere soil in the NG and the VV groups increased by 7.33% and 21.44%. The PICRUSt and FAPROTAX analysis showed that sown grass had a greater influence on the ecological function of the soil's bacteria. Cropping with *Vicia villosa* and native grass significantly increased aerobic chemoheterotrophy, nitrogen respiration, nitrate reduction related to biochemical cycles, and the relative abundance of genes related to carbohydrate metabolism and the biodegradation of xenobiotics. The bacterial network complexity in the NG group was higher than that in the CK and VV groups and was greatest in the hub nodes (OTU42, *Bacteroidota*; OTU541, *Nitrospiraceae*). In this study, the ecological benefit seen in the soil's microbial function provides support to the theory that cropping with grass (*Vicia villosa*) increases the sustainable development of a jujube orchard.

## INTRODUCTION

As an important component of commercial orchards, the jujube (*Ziziphus jujuba* Mill.) is the dominant fruit in China, accounting for 98% of the world's jujube cultivation. Modernized and sustainable agriculture has improved crop production making it more efficient. More eco-friendly and effective management of jujube orchards is needed to alleviate such problems as a site's poor conditions, serious water-fertilizer loss, and

Corresponding author
Qiong Zhang, qiong1982@163.com

unreasonable fertilization methods used under current management strategies (*Das et al., 2021*; *Wu et al., 2019*). Cropping with grass in an orchard is an effective method and it has been widely used in many regions (*Ren, Li & Yin, 2023*; *Xiang et al., 2021*; *Chen et al., 2021*). It increases vegetation coverage and also enhances the inter-root interactions, which has a significant and positive effect on soil formation and soil properties (*Huang et al., 2014*). This would provide us an exercisable and constructive insight to improve soil quality and land use efficiency by sown grass in jujube orchard.

Sown grass decreases the net energy expenditure at night and reduces the temperature differences between both the annual and daily temperature of the soil (*Hassan et al., 2022*). The variations in soil moisture content, nutrient availability, and accumulated temperature have a profound effect on the structure of the microbial communities of the soil in an orchard (*Wang et al., 2023a*; *Wang et al., 2022c*; *Ma et al., 2020b*; *Sánchez et al., 2007*).

The ecological functions of soil bacteria related to carbon (C), nitrogen (N), and sulfur (S) cycles were always significantly affected by soil bacteria secretions, which directly determine the soil nutrient bioavailability (*Zhang et al., 2023*; *Durrer et al., 2021*; *Jones et al., 2016*). Soil nutrient availability and co-species competition are enhanced under the grass management method, which may influence environmental heterogeneity and place selection pressure on soil bacterial groups in the different layers. *Jiang et al. (2019)* found that sown grass increased the richness of pathogenic bacteria such as solanacearum in an orchard. *Wei, Zeng & Tan (2021)* pointed out that cropping with grass significantly influenced soil bacterial structures and its alterations in fungal guilds due to the cropping with grass may be the main cause of improved mango fruit yields. This also changes co-occurrence networks, thus further affecting the rhizosphere process for plant nutrients (*Zhang, Vivanco & Shen, 2017*). The complex relationship is visualized as a network of microbial communities, which reveals the factors that drive microbial associations under environmental perturbations. This provides some key information on how microbial groups relate to soil function (*Banerjee, Schlaeppi & Van der Heijden, 2018*). These associations are important factors contributing to microbial community stability and they also enhance root activity leading to an increase in the plant's uptake of nutrients. Due to the microbial network's sensitivity to external interference, a change in the microbial network may appear to be a marked outcome. However, few studies on the soil bacterial OTUs network under grass management have been done, thus limiting an understanding of the soils ecological function in a jujube orchard. A comprehensive analysis of the microbial network is needed to gain more knowledge of the ecological benefits of grass cultivation in an orchard.

*Vicia villosa* Roth, an annual legume with a growing period of 230 d, is widely distributed and grown in orchard in China. The total N in the soil and alkali-resolving N fixation of *Vicia villosa* was 970 and 56.8 mg kg$^{-1}$ respectively (*Wang et al., 2023b*; *Chen et al., 2016*). Its nutrient preservation qualities and low degree of lignification in the stem were found in previous studies (*Yang et al., 2022*; *De Torres et al., 2018*). Although some information about the soil microbial composition in orchards cropped with *Vicia villosa* was preliminary reported in previous studies (*Wang et al., 2022c*; *Jiang et al., 2019*), the understanding of the co-occurrence network of the microbial community remains limited. We hypothesized

that cropping with *Vicia villosa* in a jujube orchard may improve the biochemical cycles and enhance the co-occurrence network. In this study we conducted a 2-year (2019–2020) field experiment using three cultivation patterns (cleaning tillage, cropping with native grass, and cropping with *Vicia villosa*) in an orchard. Bacterial ecological function analysis was used to characterize the response of the bacterial community to different cultivations. The objectives of this study were: (a) to detect the effects of sown grass on the soil properties and the bacterial community structure; (b) to analyze the differences in the soil's bacterial and ecological functions and the co-occurrence network properties under different cultivation patterns.

## MATERIALS AND METHODS

### Study area

The experiment was conducted at the Taidong test site (36°12′34.15′N, 117°09′35.25′E) of the Shandong Academy of Agricultural Sciences from 2019 to 2020. The study site is located in Tai'an, Shandong Province (Fig. 1A) and has a seasonal, temperate, and semi-humid monsoon climate. In this area the annual precipitation is approximately 697 mm and approximately 80% of the total annual precipitation occurs between May and September. The frost-free period is 195 days (Fig. 1B). The soil type is clay loam and properties are as following: organic matter, 9.18 g kg$^{-1}$; nitrate nitrogen, 32.56 mg kg$^{-1}$; ammonium nitrogen, 20.42 mg kg$^{-1}$; available phosphorus, 15.12 mg kg$^{-1}$; available potassium, 124.32 mg kg$^{-1}$; pH, 7.7. The jujube is a 'Jinsi 4' variety, independently bred and fully ripe in early October. It was cultivated with 1.5 m row spacing and 4 m line spacing. *Vicia villosa* seeds were collected from the Shandong Tiandi Horticultural Technology Co., LTD.

### Experimental design

This study was designed as completely randomized design (CRD), and it contained two factor (cultivation, soil sampling type) in the field experiment. In the jujube orchard, three cultivation patterns were used: (1) cleaning tillage group (CK): the jujube orchard was cleared, with no weeds in this area; (2) native grass group (NG): the species include *Digitaria sanguinalis*, *Imperata hoenigii*, *Humulus scandens*, *Ixeridium chinense*, *Echinochloa.* and *Crusgalli Beauv.* var.*crusgalli*, cut three times during the experiment and kept to a height of 15~20 cm; (3) *Vicia villosa* group (VV): the seeds of *Vicia villosa* were sown in ditches in September 2018, which were approximately 30 cm away from the main stem and the planting density was 22.5 kg hm$^{-2}$. The area of each treatment was 667 m$^2$ with three replicates. And the age of jujube was 5 year old, and fruit yield per plant and single fruit size was 12.22 kg and 10.37 g. The cultivation management, including irrigation, fertilization, and pruning density, was identical as shown in Table 1.

### Sampling analysis

In September 2020 (25 d after irrigation), a total of 30 fruits were collected from six trees in each treatment. The fruit was weighed using an electronic balance. The organic acid content was determined by NaOH titration (*Bao, 2000*) and the flavonoid content was determined by ultraviolet spectrophotometry (*Palma, Corpas & Río, 2011*). The rhizosphere and
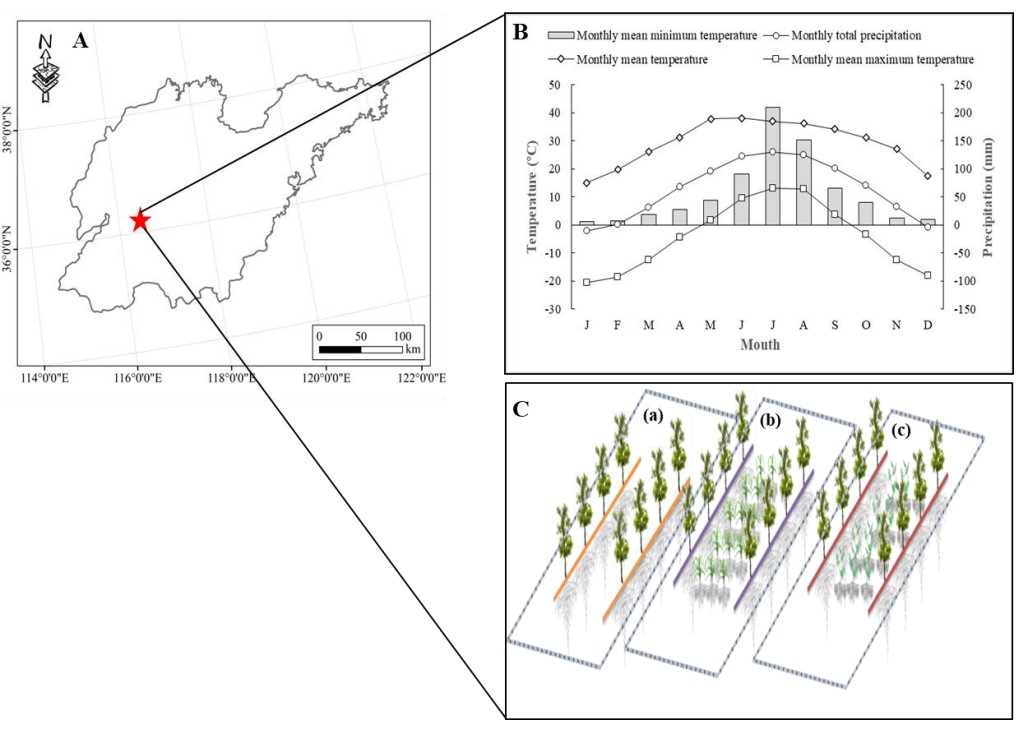

**Figure 1   The location (A) and climatic condition (B) of the field experiment.** Schematic diagram of the three cultivation patterns (C). a, b and c represent cleaning tillage (CK), native grass (NG), and *Vicia villosa* (VV) in the jujube orchard. The data of the location is provided by standard map service (http://bzdt. ch.mnr.gov.cn/).

**Table 1   The management of Irrigation, fertilizer, and pruning in the jujube orchard.**

| Year | Tree height (m) | Tree form | Prune stage | Number of secondary branches | (NH4)$_2$HPO$_4$: Urea:K$_2$SO$_4$ (kg/667 m$^2$) | Organic fertilizer (kg/667 m$^2$) | Irrigation amount (m$^3$/667 m$^2$) |
|---|---|---|---|---|---|---|---|
| 2019 | 3.5 | Spindle | Jan., Mar. | 27 | 15:20:10 | 1,000 | 180 |
| 2020 | 3.5 | Spindle | Jan., Mar. | 27 | 15:20:10 | – | 160 |

interrow soil in each cultivation was collected by root-shaking and soil auger methods at 25 d after irrigation, respectively. The rhizosphere soil was carefully collected by shaking off the soil adhering to the fine root surface (0–3 mm). The 40 × 40 × 30 cm soil samples located 30 cm away from the trunk were excavated for rhizosphere soil. The interrow soil was collected using the soil auger method and the 0–30 cm soil layer was drilled at the middle point between two rows. Each treatment gathered 0.3 kg of soil and was repeated three times to determine each index. The partial soil was numbered and placed in a refrigerator at −80 °C for later microbial sequencing. The other samples were dried and passed through a two mm sieve to measure the physicochemical properties. Soil moisture content was determined using the drying method: The covered aluminum box was dried at 105 °C, then cooled to room temperature in the dryer and weighed ($m_0$). 30∼40 g fresh

soil in the aluminum box and determine the weight ($m_1$), then were dried at 105 °C. After drying, the box was put into the dryer and cooled to room temperature for weighing. After drying for another 2 h, the weight ($m_2$) was recorded until the mass was constant. The soil moisture content was ($m_1$–$m_2$)/($m_1$–$m_0$) (*Bao, 2000*). The soil pH value was determined using a PHS-3C pH meter. Soil organic matter content (SOM) was determined by the potassium dichromate method (*Bao, 2000*). Soil total nitrogen content (TN) was determined by the Kjeldahl method. Soil available phosphorus (Olsen-P) was extracted with 0.5 mol $L^{-1}$ $NaHCO_3$ (2.5 g soil and 50 ml solution) (*Bao, 2000*).

The 16S RNA sequencing of the soil bacteria was completed by the Beijing Nuohe Zhiyuan Technology Co., LTD. Total genomic DNA of the samples was extracted by the cetyltrimethylammonium bromide (CTAB) method. The fragments in the V3+V4 region of soil bacteria 16S rRNA were amplified by general primers 338 F (5′-ACTCCT ACG GGA GGC AGCA-3) and 806 R (5′GGA CTACHV GGG TWT CTA AT-3). The products were detected by 2% AGE (agarose gel electrophoresis) and recovered by AXYGEN gel electrophoresis. A small fragment library was constructed and double-terminal sequencing was performed using the Illumina NovaSeq sequencing platform. The operational taxonomic units (OTUs) were grouped for species annotation and abundance analysis. In this study, ten nodes with the highest degree were always defined as hub nodes (*Ma et al., 2020a*). The functional analysis of FAPROTAX and PICRUSt and principal coordination analysis (PCoA) were conducted using the NovoMagic website (https://magic.novogene.com/).

### Statistical analysis

In this study, analysis of variance (ANOVA, one-way) was conducted to detect the differences in soil physicochemical properties and the relative abundance of the functional gene among the three cultivations using SPSS statistical software (version 19.0; IBM SPSS Inc., Armonk, NY, USA). A data normality test was conducted and significant differences among means were separated by the least-significant difference, (LSD) test at the $P < 0.05$ probability level. This was represented by different letters in the figures using Duncan's test. The significant differences of soil properties between rhizosphere and interrow were separated by Student's *t* test (*T* test). The SPSS statistical software was conducted to analyze the interaction of cultivation and soil type on soil properties, functional gene's relative abundance of bacterial community. The soil bacterial ecological networks of the three cultivations at the OTU level were displayed using Gephi software (The Open Graph Viz Platform, version 9.0; https://gephi.github.io/) and R software (version 4.0.3-win; *R Core Team, 2020*).

## RESULTS

### Fruit and soil physicochemical properties

Compared to the CK group, the weight of the single fruit (fruit yield per tree) in the VV and NG groups increased by 0.58 g (0.90 kg) and 0.41 g (0.65 kg), respectively (Figs. 2A and 2B). The flavonoid content was highest in the VV group, while the organic acid content in

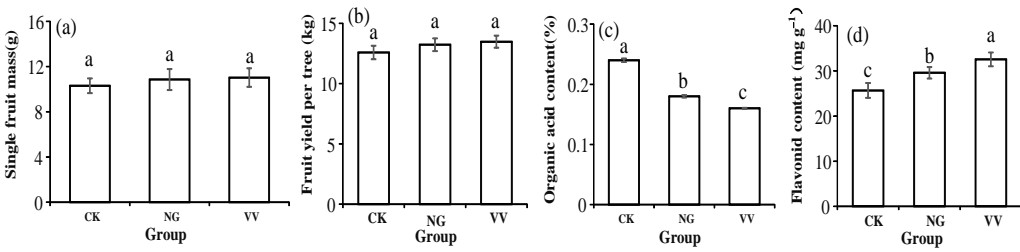

**Figure 2 The variations in single fruit mass (A), yield (B), organic acid content (C) and flavonoid content (D) under the three cultivations, respectively.** Lowercase letters on the column indicate significant differences among the three cultivation patterns.

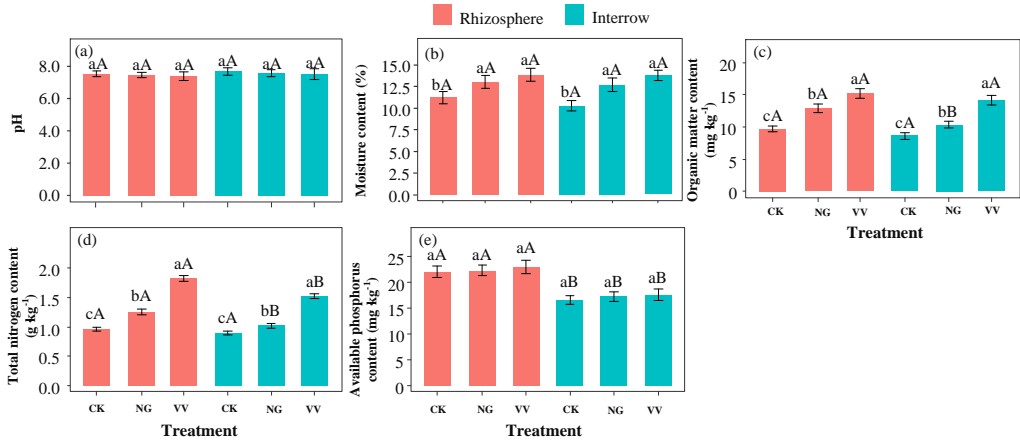

**Figure 3 The variations in soil pH (A), moisture (B), organic matter (C), total nitrogen (D) and available phosphorus (E) contents under the three cultivation patterns in the jujube orchard.** Capital letters on the column indicate the significant differences between the rhizosphere soil and the interrow soil. Lowercase letters on the column indicate significant differences among the three cultivation patterns.

the CK group was 0.08 mg g$^{-1}$ and 0.06 mg g$^{-1}$ higher than that of the VV and NG groups ($P < 0.05$, Figs. 2E and 2F).

The contents of available phosphorus and organic matter in the rhizosphere soil were higher than those in the interrow soil, which were 5.43–5.57 mg kg$^{-1}$ and 1.02–2.52 mg kg$^{-1}$, respectively. The pH in the rhizosphere soil was 0.11–0.15 lower than that found in the interrow soil (Figs. 3A, 3C and 3E). The soil moisture, total nitrogen, and organic matter in the rhizosphere in the NG and VV groups were 1.82 (2.66)%, 0.29 (0.87) g kg$^{-1}$ and 0.32 (5.55) mg kg$^{-1}$ higher than those found in the CK group (Figs. 3B and 3D). The soil pH decreased in the NG and VV groups compared to the CK group (Fig. 3A). The main effects of cultivation on mositure, organic matter and total nitrogen content were displayed, and the main effects of soil type displayed in organic matter, total nitrogen and available phosphorus. The interaction of soil type and cultivation on soil total nitrogen was significant ($P < 0.05$, Table 2).
**Table 2  Results (*F*-value) of interactions of soil type and cultivation on soil physicochemical properties.**

| Soil properties | Cultivation | Soil type | Cultivation * Soil type |
|---|---|---|---|
| pH | 0.43 | 0.91 | 0.00 |
| Mositure content | 20.46*** | 1.47 | 0.38 |
| Organic matter | 79.56*** | 17.90** | 1.90 |
| Total nitrogen | 390.49*** | 69.60*** | 8.86** |
| Available phosphorus | 0.74 | 71.67*** | 0.07 |

**Notes.**

Asterisks (*, **, ***) represent statistically significant difference $P < 0.05$, $P < 0.01$, $P < 0.001$.

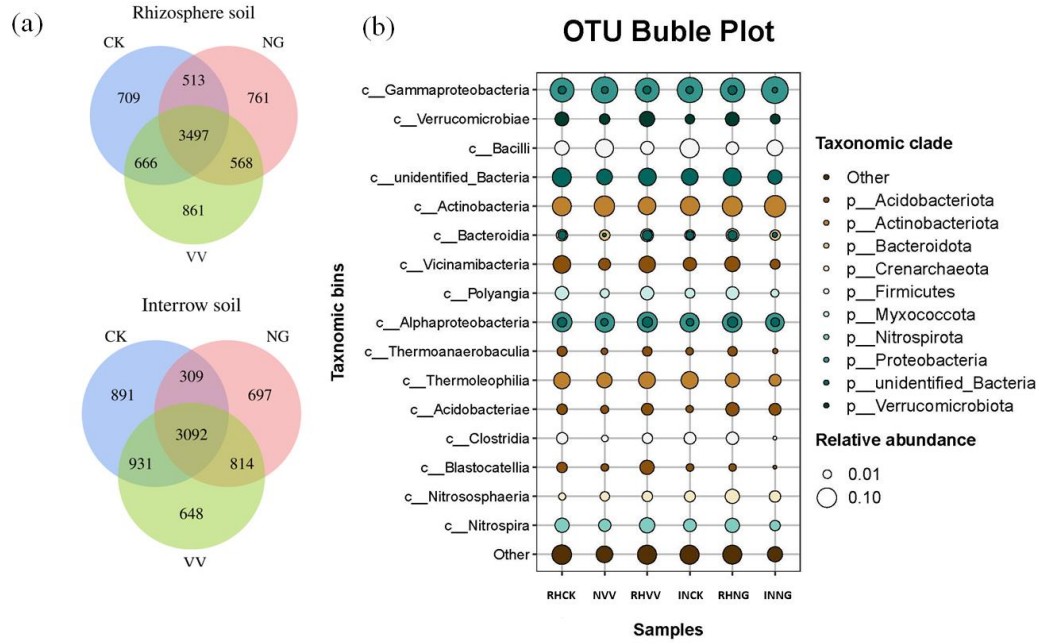

**Figure 4  Venn diagram (A) and bubble plot (B) of OTUs for bacterial sequences in the rhizosphere and interrow soil.** CK, VV and NG represent the cleaning tillage, cropping with *Vicia villosa*, and native grass in a jujube orchard, respectively.

## Bacterial diversity

A total of 4,552,560 high quality effective sequences were obtained through microbial sequencing analysis. The number of effective bases was $1.72 \times 10^9$ and the average sequence length was 376.846 bp. Compared to the CK group, the number of endemic species in the rhizosphere soil was significantly higher in the NG and VV groups, by 7.33% and 21.44% respectively, but was lower in the interrow soil (Fig. 3A). The number of Actinomycetes and Proteobacteria was higher in the NG and VV groups compared to those in the CK group, while the relative abundance of Acidobacteria and Chloroflexi was found to be the opposite (Fig. 4B).

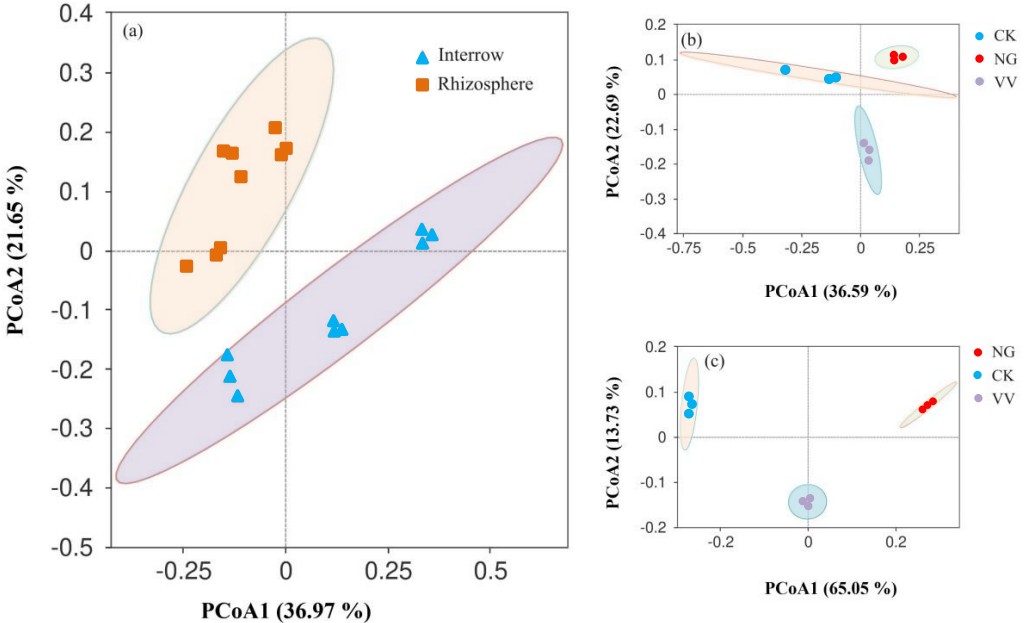

**Figure 5** **Principal coordination analysis (PCoA) of bacteria across all soil samples (A), the rhizosphere soil samples (B) and the interrow soil samples (C).** CK, VV and NG represent the cleaning tillage, cropping with *Vicia villosa*, and native grass in the jujube orchard, respectively.

PCoA analysis showed that the first variance was 36.97% and the bacteria aggregation in the rhizosphere soil was higher than that found in the interrow soil, indicating that the NG and VV groups had a greater influence on the bacterial community composition in the jujube orchard (Fig. 5A). Single PCoA analysis was performed on the interrow soil and the rhizosphere soil and the first variance was 65.05% and 36.59%, respectively, indicating that the influence on the bacterial community in the interrow soil was greater than that in the rhizosphere soil (Figs. 5B and 5C).

## Functional annotation

The PICRUSt analysis showed that the functional gene's relative abundance in translation, energy metabolism, replication and repair in the rhizosphere soil was higher than those in the interrow soil. In the interrow, it was also lowest in the NG group among the three cultivation patterns (Figs. 6D, 6E and 6G). For membrane transport, amino acid metabolism, and xenobiotics biodegradation and metabolism the relative abundance of the functional gene in the rhizosphere soil was lower than that in the interrow soil (Figs. 6A, 6B and 6J). Compared to the CK group, the VV group's relative abundance of the functional gene in poorly characterized and amino acid metabolism of the rhizosphere soil increased. The relative abundance of the functional gene in carbohydrate metabolism, xenobiotics biodegradation and metabolism, and lipid metabolism in the interrow soil increased as well ($P < 0.05$, Figs. 6F and 6I). The interaction of soil type and cultivation on replication and repair, translation metabolism of cofactors and vitamins, and lipid metabolism were significant ($P < 0.05$).

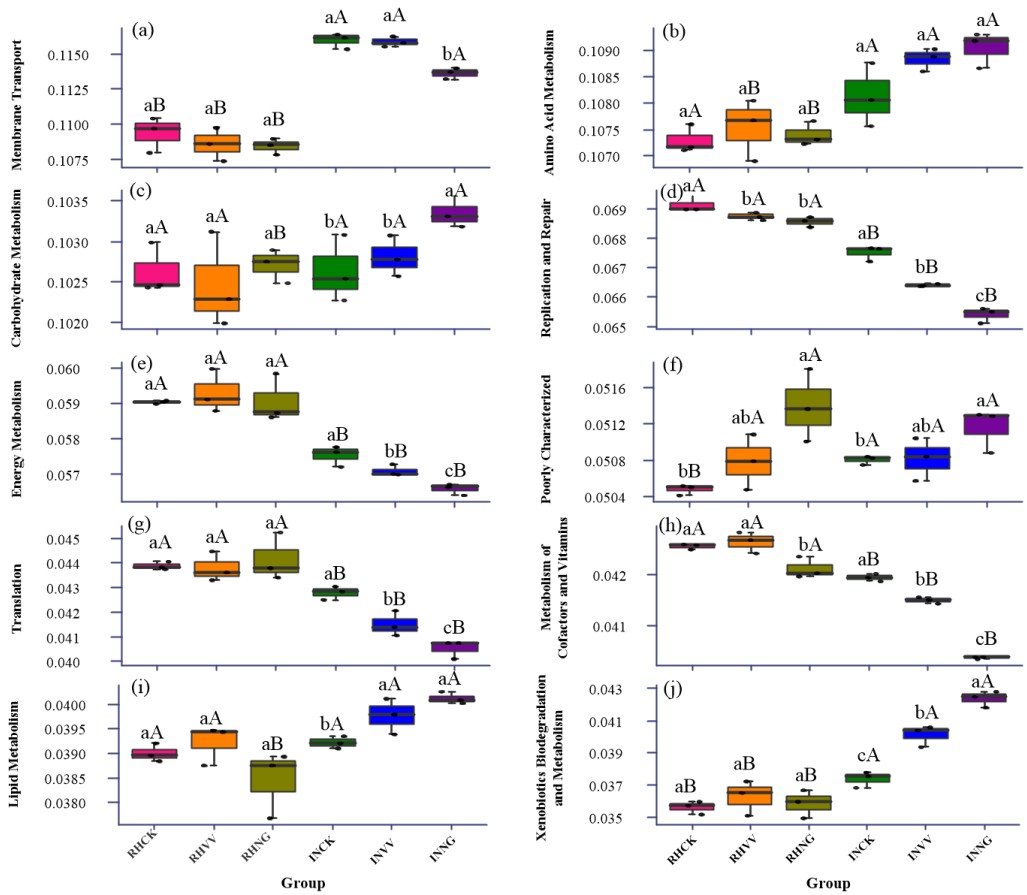

**Figure 6** **The functional analysis of bacterial communities in the rhizosphere and interrow soils under the three cultivation patterns in the jujube orchard using PICRUSt.** Capital letters indicate the significant difference between rhizosphere soil and interrow soil. Lowercase letters indicate significant differences between the three cultivation patterns. RHCK, RHVV, and RHNG represent the rhizosphere soil in the CK, VV, and NG group. INCK, INVV, and INNG represent the interrow soil in the CK, VV, and NG group.

FAPROTAX annotation showed that the functional groups in the rhizosphere soil under the three cultivation patterns were more closely associated with chemoheterotrophy, aerobic chemoheterotrophy, nitrogen respiration, and nitrate reduction than that in the interrow soil (Figs. 7A, 7B, 7E and 7G). Among the three cultivation patterns, superior nitrification and fermentation in the rhizosphere soil were found in the NG group (Figs. 7I and 7D). The association between chemoheterotrophy, aerobic chemoheterotrophy, and nitrate reduction in the VV and NG groups were enhanced compared to those in the CK group ($P < 0.05$). This association was strongest in the NG group (Figs. 7A, 7B, 7C, 7E, 7G and 7H). For main effect, functional activities displayed obviously in chemoheterotrophy, aerobic chemoheterotrophy, nitrate reduction, nitrogen respiration, nitrate respiration, aromatic compound degradation in soil type, while aerobic chemoheterotrophy, nitrogen respiration, nitrate respiration, aromatic compound degradation was in cultivation.

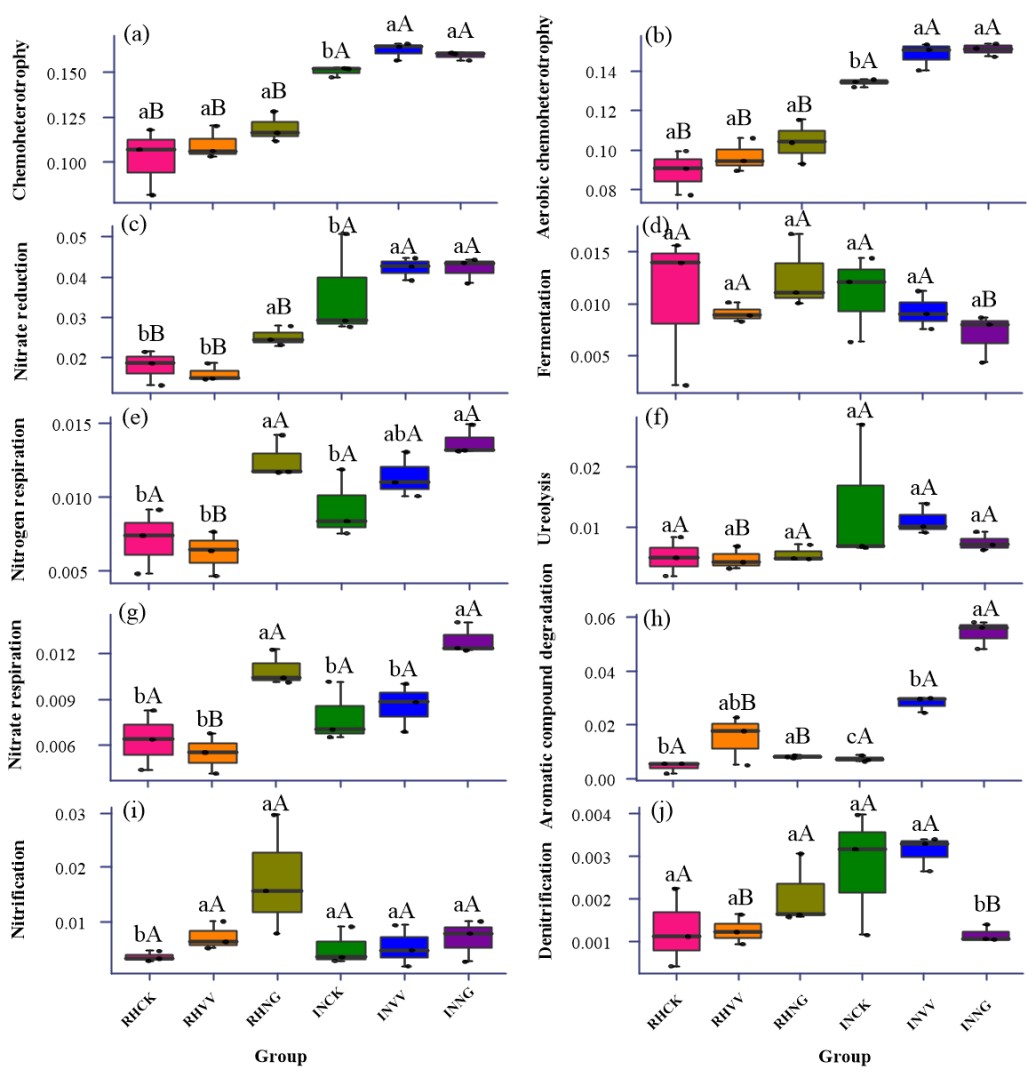

**Figure 7** The functional prediction of bacterial communities in the rhizosphere and interrow soils under the three cultivation patterns using FAPROTAX. Capital letters indicate the significant differences between rhizosphere soil and interrow soil. Lowercase letters indicate the significant differences among the three cultivation patterns. RHCK, RHVV, and RHNG represent the rhizosphere soil in the CK, VV, and NG group. INCK, INVV, and INNG represent the interrow soil in the CK, VV, and NG group.

The significant interaction on aromatic compound degradation and denitrification was displayed in Table 4 ($P < 0.05$).

## Bacterial ecological network

The complexity of the bacterial functional network in the NG and VV groups was higher than in the CK group and was highest in the NG group (Figs. 8A, 8B and 8C). For network structure, the number of nodes in the VV and NG groups was 172 and the number of nodes at the edges was 1,511 and 1,671, respectively. This was 24.77% and 37.99% higher than the number of nodes in the CK group. The majority of nodes were affiliated with
**Table 3  Results (*F*-value) of interactions of soil type and cultivation on functional gene's relative abundance of bacterial communities.**

| Functional gene's relative abundance | Cultivation | Soil type | Cultivation * Soil type |
|---|---|---|---|
| Membrane transport | 6.90[*] | 278.42[***] | 2.55 |
| Amino acid metabolism | 2.66 | 40.08[***] | 1.39 |
| Carbohydrate metabolism | 2.66 | 3.63 | 1.18 |
| Replication and Repair | 70.76[***] | 677.25[***] | 23.14[***] |
| Energy metabolism | 2.01 | 115.6[***] | 2.32 |
| Poorly characterized | 10.33[**] | 0.11 | 1.89 |
| Translation | 5.40[*] | 85.13[***] | 8.25[**] |
| Metabolism of Cofactors and Vitamins | 107.1[***] | 382.53[***] | 30.42[***] |
| Lipid metabolism | 1.64 | 20.54[**] | 6.26[*] |
| Xenobiotics Biodegradation and Metabolism | 21.55[***] | 146.79[***] | 17.11[***] |

Notes.

Asterisks (*, **, ***) represent statistically significant difference $P < 0.05$, $P < 0.01$, $P < 0.001$.

**Table 4  Results (*F*-value) of interactions of soil type and cultivation on functional activities of bacterial communities.**

| Functional activity | Cultivation | Soil type | Cultivation * Soil type |
|---|---|---|---|
| Chemoheterotrophy | 2.89 | 110.11[***] | 0.61 |
| Aerobic chemoheterotrophy | 6.05[*] | 158.71[***] | 0.27 |
| Nitrate reduction | 2.03 | 52.8[***] | 1.05 |
| Fermentation | 0.25 | 0.83 | 1.12 |
| Nitrogen respiration | 14.71[**] | 12.25[**] | 2.15 |
| Ureolysis | 0.43 | 5.18[*] | 0.60 |
| Nitrate respiration | 19.51[***] | 9.15[*] | 0.41 |
| Aromatic compound degradation | 46.52[***] | 90.85[***] | 36.33[***] |
| Nitrification | 3.59 | 2.20 | 2.16 |
| Denitrification | 0.74 | 4.40 | 5.17[*] |

Notes.

Asterisks (*, **, ***) represent statistically significant difference $P < 0.05$, $P < 0.01$, $P < 0.001$.

Proteobacteria, Actinobacteriota, and Acidobacteriota under the three treatments in the orchard. The number of nodes associated with Proteobacteria and Actinobacteriota in the VV and NG groups was higher than the number associated with Proteobacteria and Actinobacteriota in the CK group.

The degree of hub nodes and the number of generalists were higher in the VV and NG groups than those in the CK group. Most of the hub nodes belong to Acidobacteriota and Proteobacteria under the three cultivations. Some OTUs were shared as hub nodes both in the VV and NG groups, suoutas OTU42 (*Bacteroutota*), OTU1045 (*Ramlibouter*), and OTU1019 (*Streptomouts*). OTU541 inoute NG group and OTU86 in the VV group, associated with *Nitrospiraceae*, shared a higher number of hub nodes than those in the CK group.

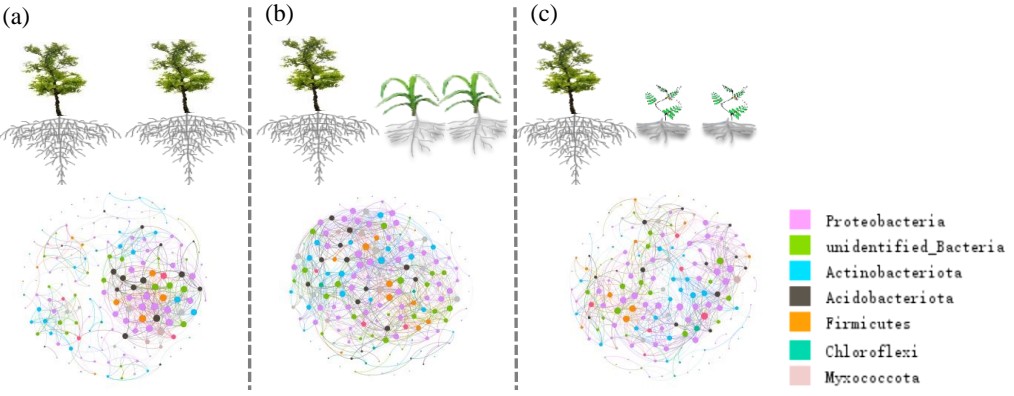

**Figure 8 The variation in the bacterial ecological networks at the out level under the three cultivation patterns: A, B and C represent the bacterial network in the CK, NG, and VV group, respectively.** A connection indicates a strong (Spearman's $P > 0.9$) and significant ($p < 0.01$) difference. Each node represents a uniqoutOTU. The nodes colored by phyla and the keystone taxa are magnified.

# DISCUSSION

Grass intercropping is an agronomic cultivation method that is widely used throughout the world due to its ecological benefits (*Zhu et al., 2022*; *Wang et al., 2022a*). In this study, the fruit biomass and flavonoid content in the CK group were lower than those in the NG and VV groups. This is mainly because sown grass promoted soil nutrient conversion and reduced fertilizer leaching, acting as a temporary reservoir of nutrients. However, the single fruit weight and fruit yield per tree is non-significant under the different cultivation, which might be caused by sample size (*Jia, He & Jin, 2018*). When the plant dies, the fine roots break down into nutrients for the jujube root system to absorb. Soil moisture, total nitrogen, and organic matter in the rhizosphere and interrow soils in the NG and VV groups were significantly higher than those in the CK group (Fig. 3). This indicated that grass cultivation in orchards is an important method that can be used to improve soil quality and promote an increased jujube yield. The results were consistent with previous studies (*Wang et al., 2022c*; *Yang et al., 2020*; *Ren et al., 2019*; *Yang et al., 2019*). We found that the soil pH decreased in the grass orchard although it was not statistically significant. This was mainly due to the root interaction that promoted root proton($H^+$) secretion (*Feng et al., 2023*; *Wang et al., 2022b*). *Liu et al. (2022)* thought that even a small pH change caused by tillage practices always determines soil bacterial diversity. The negative relationship between pH and bacteria diversity in orchard grass management was reported in *Wang et al. (2023a)*. Sown grass can increase the coverage rate of vegetation and it can reduce the evaporation of soil moisture thus enhancing the activities of the root system and microorganisms (*Tu et al., 2021*). Cropping with native grass and *Vicia villosa* in a jujube orchard also promotes root elongation, which accelerates the absorption of available nutrients. This might decrease fertilizer loss due to leaching. The residue of leaves and roots degrade in the soil and release nutrients into the soil, which might provide alternative sources of nutrition (*Wang et al., 2023d*). This greatly reduces the nutrient deficiency during key yearly growth cycles of the

jujube. The litter and root exudates contain abundant nutrients, which play a prominent role in determining the dominant population of the bacterial community.

In this study, it was found that the most abundant phyla, in all treatments, were Proteobacteria, Actinobacteria, Acidobacteria, Firmicutes, Chloroflexi, and Myxococcota (Fig. 3). Proteobacteria, Actinobacteria, and Acidobacteria are closely connected to soil nutrient availability as it relates to soil C, N, P, and S cycling (*Wang et al., 2023c*; *Xue et al., 2020*). Within the three cultivation patterns, the bacterial structure in the rhizosphere and interrow soils was markedly different. The relative abundance of Proteobacteria in the interrow soil was 30.7–45.2%, which is 7.0–18.3% higher than that found in the rhizosphere soil. This might be because the rhizosphere has direct contact with the plant surface and the soil and rhizosphere activity leads to an increase in the number of other microorganisms, especially Acidobacteria, which would decrease the relative abundance of Proteobacteria in the interrow soil. This suggests that the structure of the bacterial community in the rhizosphere soil is more sensitive to environmental disturbances compared to the bacterial community in the non-rhizosphere soil.

Cropping with *Vicia villosa* and native grass significantly increased the relative abundance of Actinomycetes (Fig. 5). With branched mycelium, Actinomycetes are a dominant group of microorganisms widely distributed in various kinds of soil. These can release extracellular hydrolase and degrade insoluble organic substances which helps the plants obtain various nutrients. Sown grass increased the soil nitrogen and SOM content, which produced an optimal ecological condition for the propagation of Actinomycetes. In general, the relative abundance of Acidobacteria is negatively correlated with soil pH and its diversity is higher when the soil pH value is about 6.0 (*Liu et al., 2016*). However, the positive correlation between the pH and the relative abundance of Acidobacteria was seen in this study. This was likely because Acidobacteria are an oligotrophic type of bacteria and the increasing nitrogen in the NG and VV groups inhibits the propagation of Acidobacteria (*Dai et al., 2018*).

Using PICRUSt and FAPROTAX analysis, the influence of chemoheterotrophy and amino acid metabolism on the bacterial function was more significant in the NG group than in the VV group. This might be because native grass management in a jujube orchard increases the vegetation composition, enhances root interaction, and promotes microbial activities in the rhizosphere soil. In addition, cropping with *Vicia villosa* and grass inhibited the number of Chloroflexi present. This was mainly due to the increase in the soil organic matter and the inhibition of the $CO_2$ fixation process. *Liu et al. (2022)* thought that the increase in soil carbon sequestration leads to the reduction of $CO_2$ emissions and higher soil organic carbon levels. Besides, as N-fixing species, cropping with *Vicia villosa* (N-fixing species) increased total nitrogen content in the rhizosphere, which would facilitate relative abundance of the functional gene in amino acid metabolism (Figs. 3 and 6).

Co-occurrence network analysis is full of insights into the complex interactions and stability of bacterial communities (*Liu et al., 2022*; *Barberan et al., 2012*). Nodes and edges in complex ecological networks can be used to classify their level of connectivity. The higher bacterial connection in an ecological network reflects higher levels of ecosystem function (*Shi et al., 2020*). The results of this study showed that the connected edges of the microbial

ecological network in the VV and NG groups were 24.77% and 37.99% higher than those found in the CK group. This indicates that cropping with *Vicia villosa* and grass produced a higher number of interactions among OTUs. The higher number of nodes and edges in the VV and NG groups' network indicated that grass management in the orchard fortified the orchard's ability to defend against environmental disturbances. The closely interlinked OTUs always share the same habitat preferences: to resist stress or deliver more nutrition at any stage of the life cycle (*Bouizgarne, Oufdou & Ouhdouch, 2014*). The NG group tended to have the greatest number of edges in the three ecological networks. This aligns with the results of the PICRUSt and FAPROTAX analysis. This implied that using more varieties of vegetation in native grass management has a remarkable and positive influence on the ecological network of soil microorganisms.

A more complicated network also slows the effects of environmental fluctuations on the whole system and stimulates an increased response in the system to allow it to cope with environmental perturbations (*Wang, Zhou & Sun, 2014*). Generalists with important topological roles are the critical cornerstones of the structural stability of the network. Hub nodes with the highest degree exert a significant influence on the microbial structure of the network (*Shi et al., 2020*; *Banerjee, Schlaeppi & Van der Heijden, 2018*; *Herren & McMahon, 2018*). In this study, the Proteobacteria, Actinobacteriota, and Acidobacteriota OTUs occupied a broad niche in the accumulation and cycling of C, N and P and were deemed to be keystone taxa (*Dang et al., 2020*; *Shi et al., 2020*). Meanwhile, some keystones (OTUs) were identified as generalists and hub nodes. The keystone taxa OTU1019 (Streptomyces) were affiliated with Proteobacteria and mainly associated with the enhancement of plant nutrient uptake (*Zhu, Tian & Li, 2019*). The OTU41 in the NG and the OTU86 in the VV were associated with Nitrospiraceae and might play an important role in the nitrification process to provide more available nitrogen. The higher yield of jujube fruit in the NG and VV groups indicates the particularly important role OTUs play in the ecological function of the system (*Wang et al., 2022c*).

## CONCLUSIONS

The soil's bacterial community structures were profoundly altered by grass cultivation in a jujube orchard. Cropping with *Vicia villosa* and native grass in the orchard had a greater effect on the ecological function of the soil bacteria. It promoted aerobic chemoheterotrophy and nitrate reduction related to biochemical cycles. Cropping with native grass increased the number of related OTUs and had a more significant effect on the functional network compared to cropping with *Vicia villosa* alone. Overall, this provides evidence that sowing grass in an orchard is beneficial in enriching the microbial community structure, improving the soil condition, and increasing the usable nutrient content for crop growth.

### Funding

This work was supported by the Taishan Scholars Program (tsqnz20231239), Natural Science Foundation of Shandong (No. ZR2023QC287) and the China Agriculture Research System (No. CARS-30-ZZ-23). The funders had no role in study design, data collection and analysis, decision to publish, or preparation of the manuscript.

### Grant Disclosures

The following grant information was disclosed by the authors:
Taishan Scholars Program: tsqnz20231239.
Natural Science Foundation of Shandong: ZR2023QC287.
China Agriculture Research System: CARS-30-ZZ-23.

### Competing Interests

The authors declare there are no competing interests.

### Author Contributions

- Shoule Wang conceived and designed the experiments, performed the experiments, prepared figures and/or tables, authored or reviewed drafts of the article, and approved the final draft.
- Zhongtang Wang conceived and designed the experiments, performed the experiments, authored or reviewed drafts of the article, and approved the final draft.
- Qing Hao performed the experiments, authored or reviewed drafts of the article, and approved the final draft.
- Bin Peng performed the experiments, prepared figures and/or tables, and approved the final draft.
- Pan Li analyzed the data, prepared figures and/or tables, and approved the final draft.
- Xuelong Qi analyzed the data, prepared figures and/or tables, authored or reviewed drafts of the article, and approved the final draft.
- Qiong Zhang conceived and designed the experiments, analyzed the data, authored or reviewed drafts of the article, and approved the final draft.

### DNA Deposition

The following information was supplied regarding the deposition of DNA sequences:
The sequences are available at Genbank SRA: PRJNA1050666.

### Data Availability

The raw measurements are available in the Supplementary File.

### Supplemental Information

Supplemental information for this article can be found online at http://dx.doi.org/10.7717/peerj.17458#supplemental-information.

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
