# Peer review of "Cropping with Vicia villosa and native grass improves soil’s bacterial structure and ecological network in a jujube orchard"

_PeerJ, doi:10.7717/peerj.17458_

## Round 0.1 · original submission · Major Revisions

Dear Dr. Wang,

Thank you for your submission to PeerJ.

Subsequent to going through the review reports, I am of the opinion that your manuscript needs an array of major and minor changes. Therefore, you are advised to carefully consider the reviewers' comments and suggestions, and make the necessary revisions to improve the quality. Specifically, you should place utmost emphasis on improving the quality of Materials and Methods section including the salient changes highlighted by the reviewers in relation to experimental design and procedure as well as the methods in data recording. Moreover, you will have to carefully address various minor suggestions on different sections of the manuscript.

It is pertinent to mention that your revised manuscript will be evaluated again to ensure that you have meticulously incorporated the necessary suggestions.

Hope to receive the revised manuscript in due course.

**Language Note:** The review process has identified that the English language must be improved. PeerJ can provide language editing services - please contact us at [email protected] for pricing (be sure to provide your manuscript number and title). Alternatively, you should make your own arrangements to improve the language quality and provide details in your response letter. – PeerJ Staff

Reviewer 1 ·

Basic reporting

The manuscript is mostly clear, with a well-organized structure conducive to understanding. However, there are instances where language could be refined to enhance clarity and readability. The authors are encouraged to proofread the manuscript meticulously or seek professional editing services to ensure the use of clear and unambiguous professional English throughout.

The authors need to revise their abstract. They should empahise on the importance and relevence of study. The hypothesis of this study is very general which needs to further refined.

Experimental design

In the experimental design section, the authors had mentioned about the fruit quality. I would like to ask :
How fruit weight and flavoniod content present in the fruit is related to their study?

Did the authors check the normality of their data before conducting statistical analyis?

Did bacterial OTU and sequences follow the normal distribution?

Validity of the findings

Novelty of this study is not clearly mentioned.

Reviewer 2 ·

Basic reporting

• English language must be improved and the introduction lacks connection between the statements.
• Explain what are the ecological functions of soil bacteria in the introduction
• Keywords need to improve like replace orchard grass, functional network, etc
• Explain about Vicia villosa, how much nutrient it fixes, life span of the species, etc
• Line 67-68: It was mentioned that few studies were done on Vicia in jujube orchards but highlight the findings of those studies.
• What were the hypothesis and research questions of the study? Mention them in the introduction chapter
• The figures are relevant, but the clarity of the figures must be improved.
• Literature well referenced & relevant

Experimental design

• Original primary research within the Scope of the journal.
• Line 80: What is the duration of the study? Is it one year or two years?. Kindly give the season of the experimentation along with the year.
• Figure 1c should give how many fertilizers applied, pruning time and intensity, tillage practices, irrigation etc should be given in table form
• Lay out of the experiment may be given separately
• What is the relevance of Figure 1d, e, f given along with a description of the experimental site
• Lines 91-97: What is the statistical design of the study? There are only three treatments and mentioned that there were three replications. It seems the replications were less to achieve enough degrees of freedom.
• Give the references for organic acid and flavonoid analysis in fruits and explain the extraction procedure.
• Line 110-114: What is the soil sampling size, and how many samples were collected in each treatment?
• Explain how the soil moisture was determined.
• Give references for each parameter studied.
• How many days after irrigation soil samples were collected for analysis of moisture and nutrients?

Validity of the findings

• The study was conducted only for one season and the findings cannot be validated without repetition.
• No growth parameters on intercrops are studied and it lacks crucial information on how much biomass they produced including root and shoot.
• Only individual fruit weight is assessed, there is no information on fruit yield per tree. This will not give a clear picture of how the fruit yield is affected due to intercropping.
• The fruit size may vary because of various operations like fruiting, number of fruits per tree, fertilizer application, moisture etc.
• Correct soil water as soil moisture throughout the manuscript
• Figure 2 X-axis labels may given as CK, VV, NG and combine both rhizosphere and inter row plots
• Periodical soil moisture and nutrients are not assessed, it is difficult to conclude how moisture and nutrients varied due to intercropping
• Replications for soil samples analysis may include since PCA is plotted only for three replications, the sample size is very small for drawing PCA.
• Discussion should be improved especially on why fruit growth, organic acids, and flavonoids are improved due to intercropping. All the results must be discussed thoroughly by interconnecting the results with suitable citations.
• Figure 7 is not readable and it must be improved for clarity
• Figure 2 DMRT used for comparison of two soils ie rhizosphere and Interspace need
recheck. They are not correctly labeled or the analysis must rechecked.
• Figure 2, 5, 6 should be modified to visualize the induvial effect of soil, intercrop and their interaction.
• Make the conclusion more constructive and objective oriented rather than generic statements

Additional comments

The study on Cropping with Vicia villosa and native grass to improve soil bacterial structure and ecological network in jujube orchard is a good attempt made by the authors. The study lacks the hypothesis and research questions. The introduction must be carefully checked and there is a lack of connection between statements. There are a few lacunas in the manuscript such as the authors saying there is improvement in fruit growth, but there is no evidence of fruit characters provided. Since the yield depends on the management practices of the orchard last year and the present year. The study was conducted only for one season. The data on periodical observations of soil moisture, nutrients and microbial diversity are most valid for confirmation of any changes due to intercropping. The methodology included is not sufficient and the background of the experimental details like the age of the orchard, cultural practices followed, any fertilizer application, frequency of irrigation etc. are missing.

Annotated reviews are not available for download in order to protect the identity of reviewers who chose to remain anonymous.

Reviewer 3 ·

Basic reporting

The article is written in good scientific language that is easy to understand. The Introduction section provides a good rationale for the choice of research topic. The links provided correspond to the material presented. The structure of the article is logical and justified by the results obtained.

Experimental design

The study aim is relevant because The article discusses the issue of the mutual influence of agro-biocenosis components to increase the productivity of gardens.
The methods presented in the work correspond to the tasks set and are described with sufficient accuracy for reproduction.

Validity of the findings

The presented results are statistically reliable, and the conclusions drawn are logically justified and beyond doubt.

---

## Round 0.2 · Major Revisions

Dear Dr. Wang

Your manuscript requires a number of Major and Minor Revisions.

You are therefore advised to consider all the suggestions and thoroughly revise and resubmit ASAP. It is important to mention that you will have to exercise utmost care in addressing these recommendations in view of the fact that your revised manuscript will undergo peer review again to examine your responses.

Reviewer 2 ·

Basic reporting

authours provoided sufficient background of research
however still connection between the statements is missing

Experimental design

Line 104: mention year instead of word previous year
Table 1: compound fertilizers mentioned need to be specify which fertilizer or mention in the N P2O5 and K20 dose
Point 14. Lines 91-97: What is the statistical design of the study? There are only three treatments and mentioned that there were three replications. It seems the replications were less to achieve enough degrees of freedom.
• Whether statically design is RBD or CRD? Clarify and mention in the text.
• Even though three replications the error degrees of freedom is less than 12. In that case experimental design and sample size wrong.
• Again in the text 2nd factor is not mentioned
Point 17. Explain how the soil moisture was determined.
• Explanation should be mentioned in the text along with reference.
Point 19. How many days after irrigation soil samples were collected for analysis of moisture and nutrients? Response 19: Thank you very much for your suggestions. We collected soil samples at 25 d after irrigation, and we have added the timing in the revision.
 You said it is mentioned in the revised text. But I could not find the included text. No reference for moisture estimation, whether its is gravimetric or volumetric estimation, and not included any timing in the methodology.
 Methodologies still not explained properly

Validity of the findings

Figure 3: as capital letter indicates difference between rhizopehre and interrow the DMRT used seems wrong in Figure 3c and d.
 Figure 3c In interrow bar for NG it is mentioned B and for CK it is mentioned A.
 similarly in figure 3d in rhizophere data eventhough enough differences seen data it is mentioned A for all the bars and in interrow data for lowest value in CK treatment it is mentioned A and for NF and VV it is mentioned that BB. This not the proper way of comparison.
From Review 1: Point 23. The fruit size may vary because of various operations like fruiting, number of fruits per tree, fertilizer application, moisture etc. Response 23: Yes, the fruit size may vary because of various operations. In this study, the operations were consistent in three treatments, so the differences in the fruit size were caused by the different treatment in jujube orchard, and these consistent operations were had a description in the revised manuscript.
In the case of consistent operations in orchard and uniformity of trees what was the age of the trees, and give the CV of fruit yield and fruit size before experiment initiation.
 Figure 6: d, h, j shows significant difference in interrow soils, but in these figures DMRT shows no difference. Statistical analysis needs rechecking.

From Review round 1: Point 30. Figure 2 DMRT used for comparison of two soils ie rhizosphere and Interspace need recheck. They are not correctly labeled or the analysis must rechecked. Response 30: Thank you very much for pointing out the mistakes. We have rechecked and modified Figure 2 and the analysis in the revised manuscript.
Figure 2 now it is figure 3. The same mistake if found and authors did not corrected the mistake.
 Reason for non-significant difference due to cropping systems needs to be explained. The single fruit weight and fruit yield per tree is non-significant in all the treatments
 Way of presenting the two way data is wrong. Authous required to give factor 1 and factor 2 separately and then interaction effect for clarity and self explanatory data.

Additional comments

there significant flaws seen in the data and DMRT shown in the bar graphs. authours need to be rechecked for DMRT. the same comments made in reviison 1 also. I suggest authours to present data acoording to two way experiment design including factor 1, 2 separatly along with interaction data to avoid mistakes in the presentations.

---

## Round 0.3 · accepted · Accept

Dear Dr. Wang,

Thank you for your submission to PeerJ.

I am writing to inform you that your manuscript - Cropping with Vicia villosa and native grass improves soil's bacterial structure and ecological network in a jujube orchard - has been Accepted for publication.

Congratulations!

Reviewer 2 ·

Basic reporting

well written and acceptable

Experimental design

methods described in detail

Validity of the findings

accepted all the corrections made. conclusions are well stated with justifiable results and discussion.

Additional comments

minor corrections for gramatical mistakes may be adressed